# Broad Transcriptomic Impact of Sorafenib and Its Relation to the Antitumoral Properties in Liver Cancer Cells

**DOI:** 10.3390/cancers14051204

**Published:** 2022-02-25

**Authors:** Laura Contreras, Alfonso Rodríguez-Gil, Jordi Muntané, Jesús de la Cruz

**Affiliations:** 1Instituto de Biomedicina de Sevilla, Hospital Universitario Virgen del Rocío/CSIC/Universidad de Sevilla, E-41013 Seville, Spain; lcontreras@us.es (L.C.); arg@us.es (A.R.-G.); 2Departamento de Genética, Facultad de Biología, Universidad de Sevilla, E-41012 Seville, Spain; 3Centro de Investigación Biomédica en Red de Cáncer (CIBERONC), E-28029 Madrid, Spain; 4Departamento de Fisiología Médica y Biofísica, Universidad de Sevilla, E-41009 Sevilla, Spain; 5Centro de Investigación Biomédica en Red de Enfermedades Hepáticas y Digestivas (CIBEREHD), E-28029 Madrid, Spain

**Keywords:** hepatocellular carcinoma cell line, gene ontology, RNA-Seq, RNA synthesis, Sorafenib

## Abstract

**Simple Summary:**

Hepatocellular carcinoma (HCC) is the fourth most frequent cause of cancer-related mortality worldwide. While ablation, resection and orthotopic liver transplantation are indicated at an early stage of the disease, Sorafenib (Sfb) is the current most administrated first-line treatment for advanced HCC, even though its therapeutic benefit is limited due to the appearance of resistance. Deep knowledge on the molecular consequences of Sfb-treatment is essentially required for optimizing novel therapeutic strategies to improve the outcomes for patients with advanced HCC. In this study, we analyzed differential gene expression changes in two well characterized liver cancer cell lines upon a Sfb-treatment, demonstrating that both lines responded similarly to the treatment. Our results provide valuable information on the molecular action of Sfb on diverse cellular fundamental processes such as DNA repair, translation and proteostasis and identify rationalization issues that could provide a different therapeutic perspective to Sfb.

**Abstract:**

Hepatocellular carcinoma (HCC) is one of the most frequent and essentially incurable cancers in its advanced stages. The tyrosine kinase inhibitor Sorafenib (Sfb) remains the globally accepted treatment for advanced HCC. However, the extent of its therapeutic benefit is limited. Sfb exerts antitumor activity through its cytotoxic, anti-proliferative and pro-apoptotic roles in HCC cells. To better understand the molecular mechanisms underlying these effects, we used RNA sequencing to generate comprehensive transcriptome profiles of HepG2 and SNU423, hepatoblastoma- (HB) and HCC-derived cell lines, respectively, following a Sfb treatment at a pharmacological dose. This resulted in similar alterations of gene expression in both cell lines. Genes functionally related to membrane trafficking, stress-responsible and unfolded protein responses, circadian clock and activation of apoptosis were predominantly upregulated, while genes involved in cell growth and cycle, DNA replication and repair, ribosome biogenesis, translation initiation and proteostasis were downregulated. Our results suggest that Sfb causes primary effects on cellular stress that lead to upregulation of selective responses to compensate for its negative effect and restore homeostasis. No significant differences were found specifically affecting each cell line, indicating the robustness of the Sfb mechanism of action despite the heterogeneity of liver cancer. We discuss our results on terms of providing rationalization for possible strategies to improve Sfb clinical outcomes.

## 1. Introduction

Hepatocellular carcinoma (HCC) is the most common type of primary liver cancer in adults [1]. HCC has been described as the sixth most common neoplasia and the fourth most frequent cause of cancer-related mortality in men and women worldwide, being the main cause of death in people with cirrhosis (see [2] and references therein). The prognosis of HCC is strictly determined by the staging of the tumor and the hepatic function of the patients. The Barcelona-Clinic Liver Cancer Classification (BCLC) is currently a useful method for staging patients and recommending treatments depending on the number and size of the tumor nodules, liver function, the presence of vascular invasion, and extrahepatic metastasis [3,4]. While ablation, resection, and orthotopic liver transplantation are indicated at very early and early stages of the disease [5,6], Sfb is the current most administered drug for the treatment of patients advanced HCC [6,7,8]. Unfortunately, Sfb only provides very limited clinical benefit to patients, the five-year survival rate remaining extremely low [7,8]. Thus, there is a need to develop novel therapies that alone or in combination with Sfb could improve the outcomes for patients with advanced HCC [2,6,9].

Sfb is a multiple tyrosine kinase receptor inhibitor such as the vascular endothelial growth factor receptors 2 and 3 (VEGFR2 and VEGFR3), the platelet-derived growth factor receptor beta (PDGFR-ß), FLT3 and c-KIT, as well as the Raf kinases, which are integral components of the Ras/Raf/mitogen-activated protein (MAP)/extracellular signal-regulated kinase (ERK) kinase (MEK)/ERK signaling cascade [10,11]; thus, Sfb also downregulates the phosphoinositide-3 kinase (PI3K)/Akt/mammalian target of rapamycin (mTOR) that regulates fundamental cellular processes [12]. Sfb exerts a potent anti-proliferative and pro-apoptotic activity against HCC cells [13,14]. We and others have shown that this is likely due to the generation of a potent endoplasmic reticulum (ER) stress that leads to the sequential induction of autophagy and apoptosis processes in hepatoblastoma (HB) and HCC cell lines and tumor-derived xenograft mice models [13,15]. However, many of the molecular events by which Sfb exerts its antitumor activity remain unclear.

Previous studies have examined the expression level of numerous transcripts in selected HB and HCC cell lines, which differ in their genetic characteristics, treated with Sfb using DNA microarray technology [10]. The aim of the present study was to further determine the impact of Sfb in HB and HCC cells using high-throughput RNA-Seq technology. The study was performed in two different liver cancer cell lines differing in differentiation stage and the expression of p53: HepG2 (well-differentiated HB cell line; wild-type p53) and SNU423 (moderately differentiated HCC cell line; in-frame p53 gene truncation of amino acids 126–132) [16]. Herein, we infer, from our RNA-Seq analysis, the specific pathways that seem to be activated and inhibited upon Sfb treatment and compare the obtained results with previously published ones obtained using microarray analysis of gene expression. Both, similarities and differences in the transcriptome with and without Sfb in the two cells lines were discussed.

## 2. Materials and Methods

### 2.1. Cell Lines, Culture Conditions, and Sorafenib Treatment

HepG2 and SNU423 cell lines were obtained from the American Type Culture Collection (ATCC/LGC Standards, S.L.U., Barcelona, Spain). As described earlier, cell lines were selected according to their origin, cellular differentiation stage, and p53 genetic status: HepG2 (wild-type p53. HB-8065™) [17] and SNU423 (in-frame p53 gene deletion. CRL-2238) [18]. Both cell lines were negative for mycoplasma contamination. Cells were cultured in minimal essential medium (MEM) with Earle’s balanced salts with L-glutamine (ref E15–825, PAA Laboratories Inc., Toronto, ON, Canada) supplemented with 10% fetal bovine serum (FBS, F7524, Sigma Aldrich, Lot No. 022M3395, endotoxin < 0.2 EU/mL), 1% sodium pyruvate (ref. S11–003, PAA Laboratories Inc.), 1% non-essential amino acids (ref. M11–003, PAA Laboratories Inc., Toronto, Canada) and penicillin–streptomycin solution (100 U/mL–100 µg/mL; ref. P11–010, PAA Laboratories Inc., Toronto, Canada); cells were grown in culture flasks at 37 °C in a humidified incubator with 5% CO_2_ until reaching a density of 100,000 cells/cm^2^. Sfb was added at the concentration of 10 µM at 24 h after plating, and lysates were obtained after 12 h treatment. Sorafenib (Sfb, ref. FS10808; Carbosynth Ltd., Compton, UK) was dissolved in dimethyl sulfoxide (DMSO) as a stock solution (100 mM).

### 2.2. Cell Proliferation

The measurement of bromodeoxyuridine (BrdU) incorporation was completed as exactly described in [13] using a commercial kit (ref. 11 647229001, Roche Diagnostics, Mannheim, Germany). Cells were seeded in 96-well plates at low density (15,000 cells/cm^2^). Two hours before cell harvesting, 20 μL of 10 μM BrdU was added to the cultures. DNA was denaturalized with 200 μL FixDenat solution included in the commercial kit for 30 min at room temperature. After removal, cells were incubated with 100 μL of monoclonal anti-BrdU antibody HRP conjugated for 90 min at room temperature. Afterwards, cells were washed with PBS buffer (137 mM NaCl, 2.7 mM KCl, 10 mM Na_2_HPO_4_, 1.8 mM KH_2_PO_4_, pH 7.4) and incubated with 100 μL revealing solution for 15 min at room temperature. Absorbance at 370 nm (A_370_) was measured using an Infinite 200 PRO microplate reader (Tecan, Männedorf, Switzerland). Cell cycle progression was assessed by flow cytometric analysis. For this, cells were seeded 24 h before treatment to a final confluence of 70% in a 6-well plate. Afterwards, cells were treated with 10 µM Sfb for 12 h and then harvested and fixed in 70% ethanol in PBS buffer overnight at 4 °C. After this, cells were resuspended in PBS buffer and incubated with 0.5 mg/mL RNase A during 1 h at 37 °C. Propidium Iodine (PI) was added to a final concentration of 0.05 mg/mL for 20 min at room temperature. Finally, cells were filtered to avoid aggregation and cell cycle progression was assessed using a FACSCantoTM Flow Cytometer and analyzed using the FACSDiva software (BD Biosciences, Allschwil, Switzerland).

### 2.3. RNA Extraction

Total RNA was extracted from each sample using a RNeasy mini kit according to the manufacturer’s instructions (QIAGEN, Hilden, Germany). RNA from each sample was then stored at −80 °C before further analyses.

### 2.4. mRNA Library Preparation

Libraries were prepared using RNA of cells treated either with the vehicle or with Sfb. Total RNA was prepared as described above. Then, concentration and quality of the RNA were assessed with Qubit (Qubit™ DNA HS assay, Thermo Fisher Scientific, Waltham, MA, USA) and a 2100 Bioanalyzed Nano Chip (Agilent Technologies Genomics, Santa Clara, CA, USA), respectively. RNA Integrity Number (RIN) values were > 9 in all RNA samples. Polyadenylated RNA was isolated from the total RNA using NEBNext Oligo d(T)_25_ beads (New England Biolabs, Ipswich, MA, USA) according to the manufacturer’s instructions. Samples were normalized to an equivalent concentration of 67.3 ng/µL and prepared for RNA Ilumina Sequencing. For HepG2 cell lines, three biological replicates were obtained, while for SNU423 cells only two biological replicates were collected.

### 2.5. RNA Sequencing and Data Analyses

RNA sequencing was performed with the NextSeq500 Mid-Output and 2 × 75 pb length parameters (paired-end). RNA-Seq data were first filtered using the FASTQ Toolkit v1.0.0 program and then analyzed using the BaseSpace Onsite v3.22.91.158 from Illumina. Only genes that were upregulated or downregulated with a *p*-value < 0.5 and [log_2_(fold changes)] ≥±0.5 were selected. Data presented in this study has been submitted to the Gene Expression Omnibus database under the accession number GSE186280.

### 2.6. Gene Set Enrichment Analysis and Over-Representation Analysis

To identify functional categories significantly affected by the Sfb treatment, Gene Set Enrichment Analysis (GSEA) [19,20] and Over-Representation Analysis (ORA) were performed using the Reactome database (https://reactome.org; last accessed 3 February 2022) [21] downloaded from the Molecular Signatures Database v. 7.1 (https://www.gsea-msigdb.org; last accessed 3 February 2022). For GSEA, genes were ranked according to the *p*-value and the direction of change (up or downregulated) obtained from the RNA-Seq data. For the combined analysis of both cell lines, the product of the *p*-value of each cell line was used for the ranking. GSEA was performed using the GSEA software with the default parameters for pre-ranked lists. ORA was performed using a Fisher’s exact test [22], selecting the genes with a *p*-value lower than 0.001 in the RNA-Seq data, either in each cell line or in both cell lines.

### 2.7. Real-Time Quantitative PCR

To confirm mRNA-Seq results, real-time quantitative PCR (RT-qPCR) was performed on a set of selected genes. Total RNA was obtained as previously described and RNA samples were treated with DNase I (Promega, Madison, WI, USA) to remove all traces of DNA. After this treatment, DNase I was inactivated by incubation of the samples at 65 °C for 10 min. RNA was then reverse transcribed using SuperScript™ III First Strand Synthesis for RT-PCR according to the manufacturer’s instruction (Invitrogen, Walthman, MA, USA) and random hexamer primers (Roche, Switzerland). Each reaction contained 250 ng of RNA in a total volume of 25 μL. RT-qPCR was performed using SYBR^®^ Green Premix Ex Taq™ 2X (Takara, Japan) and primer specifics of each transcript. The ribosomal RNA 28S (28S rRNA) was used as an internal normalization control.

Primers pair used for the RT-qPCR were as follows: β-actin mRNA: 5′TCCCTGGAGAAGAGCTACGA3′ (forward) and 5′AGGAAGGAAGGCTGGAAGAG3′ (reverse); *BAX* mRNA: 5′TCCACCAAGAAGCTGAGCGAG3′ (forward) and 5′GTCCAGCCCATGATGGTTCT3′ (reverse); *BIM* mRNA: 5′CACCAGCACCATAGAAGAA3′ (forward) and 5′ATAAGGAGCAGGCACAGA3′ (reverse); *BIRC3* mRNA 5′ATGCTTCTGTTGTGGCCTGA3′ (forward) and 5′ACTCTGAACGAATCTGCAGCT3′ (reverse); *BOP1* mRNA: 5′CTGATTCACCAGCTGAGCC3′ (forward) and 5′GACGCCACCAACAGGAAG3′ (reverse); *CEBPβ* mRNA: 5′AAACTCTCTGCTTCTCCCTCTGC3′ (forward) and 5′CTGACAGTTACACGTGGGTTGC3′ (reverse); *CPEB4* mRNA 5′CACCAACACCCTCCTCTTCC3′ (forward) and 5′TTCAGGGGCGTTATTCCACC3′ (reverse); *DUSP1* mRNA: 5′CCTGAGTACTAGCGTCCCTG3′ (forward) and 5′CAGGTACAGAAAGGGCAGGA3′ (reverse); *EIF4E2* mRNA: 5′TGAAAGATGATGACAGTGGGGA3′ (forward) and 5′CTGATTCTTGTCTCGTTCCGT3′ (reverse); *EPOP* mRNA 5′AGTTTTCGGGGTGACAGTCC3′ (forward) and 5′AGATGGAAGGAGGCAGGGAT3′ (reverse); *FEN1* 5′TGGGGTCAAGAGGCTGAGTA3′ (forward) and 5′GTGGATCCCTTGGGTTCTGG3′ (reverse); *GADD45B* 5′TGGGAAGGTTTTGGGCTCTC3′ (forward) and 5′TCCAGCGTCATGTTGCAATTATA3′ (reverse); *IDI1* 5′AACCACCTCGACAAGCAACA3′ (forward) and 5′TGTTCTCGTTCAGGTGACAA3′ (reverse); *PCNA* 5′AAAGTCCAAAGTCAGATCTGGTC3′ (forward) and 5′ACTGCATTTAGAGTCAAGACCCT3′ (reverse); *PHB* mRNA: 5′TCAACATCACACTGCGCATC3′ (forward) and 5′ATAGTCCTCTCCGATGCTGG3′ (reverse); *SMAD7* mRNA: 5′CCCCTCCTCTCCCTCATCAA3′ (forward) and 5′GGCTGGCAGGAAGGGAATAA3′ (reverse); *TPI* mRNA: 5′GGACTCGGAGTAATCGCCTG3′ (forward) and 5′TGTTGGGGTGTTGCAGTCTT3′ (reverse); *VEGFA* 5′CCATCCAATCGAGACCCTGG3′ (forward) and 5′CTCCAGGCCCTCGTCATTG3′ (reverse); 28S rRNA: 5′CAAAGCGGGTGGTAAACTCC3′ (forward) and 5′TTCACGCCCTCTTGAACTCT3′ (reverse).

### 2.8. Western Blot Analysis

Protein extracts were obtained by lysing cell pellets at 100 °C for 10 min in 2× Laemmli buffer (125 mM HCl-Tris, pH 6.8, 4% SDS, 0.02% bromophenol blue, 20% glycerol, 200 mM DTT). Cellular extracts were then sonicated in a Bioruptor (Diagenode, Liège, Belgium) for 1 min at high intensity. Protein extracts were subjected to 10% SDS-PAGE and transferred to nitrocellulose membranes (Amersham^TM^ Protran^®^ 0.45 µm, GE Healthcare Chicago, IL, USA). The membranes were blocked for 1 h with 5% skim milk in TTBS (15 mM HCl-Tris, pH 7.5, 200 mM NaCl, 0.2 M NaCl, 0.1% (*v*/*v*) Tween-20), followed by incubation with primary anti-NDUFS1 (Abcam, Cambridge, UK, ab169540, 1:10,000 dilution), anti-NDUFS2 (Abcam, ab103024, 1:2000 dilution), anti-NDUFV2 (Abcam, ab183715, 1:2000 dilution) and anti-GAPDH (Santa Cruz, sc-47724, 1:1000 dilution) at 4 °C overnight.

After washing with TTBS buffer, the membranes were incubated with HRP-conjugated secondary antibody (Bio-Rad, Hercules, CA, USA) for 1 h at a 1:5000 dilution at room temperature. Proteins were detected using an enhanced chemiluminescence detection kit (Pierce^TM^ Super-Signal West Pico, Thermo Fisher Scientific, Waltham, MA, USA) in a ChemiDoc^TM^ Touch Imaging System (Bio-Rad) and the relative intensity value quantified with the Image Lab software provided with this system.

### 2.9. Statistical Analyses

Statistical analyses were performed with the Prism 6.01 software (GraphPad, San Diego, CA, USA). Data were generated from several repeats (at least three) of different biological replicates (at least three). Mean ± S.D. were represented in the different graphs. To determine significance, Student’s tests for unpaired samples with confidence intervals of 95% were computed. Significance between conditions were indicated with the symbols * *p* < 0.05, ** *p* < 0.01, *** *p* < 0.001, and **** *p* < 0.0001. Regression plots and determination of Pearson coefficients and *p*-value were performed using the R software (Institute for Statistics and Mathematics, Vienna, Austria). A Venn diagram was computed using EulerAPE software [23].

## 3. Results and Discussion

### 3.1. Transcriptional Changes Caused by Sorafenib in Hepg2 Hepatoblastoma and SNU423 Hepatocellular Carcinoma Cell Lines

Sfb is an inhibitor of several kinases involved in tumor cell proliferation and angiogenesis, including Raf, VEGFR and PDGFR [10,24]. Nowadays, Sfb is one of the most used molecular targeted drugs for the treatment of advanced inoperable HCC with significant but unfortunately modest anticancer results [7,8]. To better understand the biological consequences of the treatment, two different model liver cancer cell lines (HepG2 and SNU423 cell lines) were treated with Sfb 10 µM for 12 h and their transcriptomes analyzed by RNA-Seq and compared to those of untreated cells grown in the same conditions. It has been previously shown that at this concentration, Sfb induces apoptosis and significantly reduces cell proliferation in cultures of the two liver cancer cell lines [16].

Principal component analyses demonstrated strong consistency between repeats of each sample and clearly separated data from treated vs. untreated cells in both cell lines (data not shown). Numerous changes were observed in both cell lines when treated with Sfb (see Appendix A). When we took into consideration changes in gene expression with an established *p*-value lower than 0.001, our RNA-Seq analysis identified 2140 and 1347 differentially expressed genes (DEGs) upon Sfb treatment in HepG2 and SNU423 cells, respectively. From these, 1166 and 677 were upregulated while 974 and 670 were downregulated, respectively, Appendix A displays the list of DEGs (*p*-value < 0.001) and highlights those genes with log_2_(FC) either higher than 1.5 or lower than −1.5 in Sfb-treated HepG2 (265 vs. 298 genes) and SNU423 (264 vs. 175 genes) cells, respectively. All these results are summarized in Figure 1A.

To verify whether the RNA-Seq results were valid, we first checked a set of genes using RT-qPCR, chosen in a random way. In HepG2 cells, we checked *ACTB* (β-actin), *BIRC3*, which encodes an inhibitor of apoptosis, *CEBPβ*, which encodes for a leucine-zipper transcriptional factor regulator of, among others, genes involved in the immune and inflammatory responses, *CPEB4*, a RNA binding protein that regulates activation of UPR, *DUSP1*, which encodes for the Dual Specificity Protein Phosphatase 1, *GADD45B*, the growth arrest and DNA damage-inducible protein, *SMAD7*, which encodes for a nuclear protein involved in the inhibition of the TGF-beta receptor and *VEGFA*, the vascular endothelial growth factor A as genes showing a positive log_2_(FC), and *BAX*, a BCL2 family member with a role as a mitochondrial apoptotic activator, the gene encoding the translation initiation factor eIF4E2, *EPOP* encoding for an scaffold protein, *FEN1* encoding for a DNA nuclease, *IDI1* encoding for a peroxisomal enzyme, *PCNA*, which encodes for a cofactor of the replicative DNA polymerase and the gene for the triosephosphate isomerase (TPI) as genes showing a negative log_2_(FC). In all cases, a significant and similar trend was obtained in the levels expression of the genes tested in Sfb-treated versus non-treated cells by RT-qPCR or RNA-Seq analysis (Appendix A); thus, our RT-qPCR analysis was consistent with the obtained RNA-Seq data. In SNU423 cells, we checked *BIM*, which encodes for a proapoptotic protein, *BOP1* encoding for a factor involved in the assembly of 60S ribosomal subunits *BIRC3*, *CPEB4*, *DUSP1*, *GADD45B*, and *SMAD7* as gene showing a positive log_2_(FC), and eIF4E2, *EPOP*, *IDI1*, *PHB* (Prohibitin), which encodes for a protein proposed to play an antiproliferative role, *PCNA* and *TPI* as genes showing a negative log_2_(FC). Similarly, gene expression levels obtained using RNA-Seq and RT-qPCR methods were in good agreement each other with only small variation in the magnitude of expression (Appendix A).

Interestingly, when the RNA-Seq data were globally analyzed, a strong overlap between the set of genes misregulated by Sfb in HepG2 and SNU423 cells was observed. Indeed, changes in gene expression altered by Sfb treatment were highly positively correlated in both cell lines (Figure 1B). The linear regression for the common DEG genes shows a very high correlation (y = 1.055 × −0.0158), with a value of the Pearson Correlation coefficient as high as *r* = 0.9294. This result indicates that despite differences among the set of DEGs in each liver cancer cell line, the overall response to the Sfb treatment is very similar between both cell lines. Moreover, no special features regarding gene, ORF, 5′UTR or 3′UTR length, or GC content were found among the DEGs upon a Sfb treatment (data not shown).

Focusing on HepG2 cells, which correspond to the cell line routinely studied in our laboratory, when genes are analyzed individually (see Figure 2A), those showing the highest positive log_2_(FC) with the lowest *p*-value were mainly stress-response genes such as *INHBE*, *ATF3*, *NUPR1*, *PPP1R13B*, *PPP1R15A* (also known as *GADD34*), *DDIT3*, *DUSP1*, *DUSP8*, or *ERN1*, as well as genes such as *PDGFA* (Platelet Derived Growth Factor subunit A) and *VEGFA* (Vascular Endothelial Growth Factor A), which correspond to growth factors that activate the PDGFR and VEGFR receptors, respectively, in charge of transducing their extracellular signals into the cell and described as direct targets of Sfb [10,24]. Their upregulation by Sfb might be related to secondary cell response as a consequence of the inhibition of PDGFR and VEGFR receptors. Consistently, most abovementioned genes regulate important cellular processes leading to inhibition of cell growth and proliferation and/or induction of apoptosis. As examples, *INHBE* is a member of the TGF-beta (transforming growth factor-beta) superfamily of protein genes, which encodes a preproprotein that requires proteolytical processing to generate the inhibin beta E subunit. Inhibins downregulate different cellular processes, among them cell proliferation and apoptosis [25,26]. In agreement with our data, it has been reported that *INHBE* is upregulated under conditions of endoplasmic reticulum (ER) stress, in which it is involved in the Sfb signaling [13,27]. In addition, DUSPs dephosphorylate many key signaling molecules, including MAPKs, leading to the reduction in the duration, magnitude, or spatiotemporal profiles of the activities of MAPKs [28]. *ATF3*, a gene for a transcriptional factor that functions in general adaptive responses and whose expression is induced by various stimuli including ER stress [29], is also upregulated upon the Sfb treatment. NUPR1 is a transcription regulator that induces autophagy and apoptosis through upregulation of ER stress-related factors including DDIT3 [30]. DDIT3, also known as CHOP or GADD153, is a central transcriptional factor, also induced by ER stress, that triggers apoptosis through inhibition of BCL2 and upregulation of BIM, which regulate BAX-BAK-mediated mitochondrial outer membrane permeabilization [31,32]. We have also previously shown that Sfb induced a sustained and progressive increase in CHOP expression by Western blot analysis [13]. Finally, it is worth mentioning another example of an upregulated gene that is *ERN1*, which encodes the transmembrane kinase IRE1, which functions as a general sensor of unfolded proteins during the activation of the unfolded protein response (UPR) upon ER stress [33]. Additional DEGs showing upregulation corresponded to *RAB42* and *EMMOD2* related to GTP-binding or GTPase activation signaling transduction or *FBXO9* and *TRIM56* related to a ubiquitin ligase function. On the other hand, individual genes showing both lowest negative log_2_(FC) and *p*-value corresponded to chaperones, and co-chaperone proteins that alleviate the tendency of pre-existing proteins from aggregation or help the folding of nascent ones [34]. Thus, *HSPA1A*, *HSPA8*, *DNAJA1* and *HSPA1B* are among the genes showing the most significant downregulation (Figure 2A). This result is, however, paradoxical since the ER stress caused by Sfb leads to the activation of UPR, thus, requiring further clarification. Similar results to these found for HepG2 cells were also uncovered for SNU423 cells when treated with Sfb (Figure 2B), indicating the robustness of the cellular responses to this drug.

Next, we performed enrichment analyses of differentially expressed genes using the REACTOME database to identify the biological significance of the genes affected by the Sfb treatment common to both liver cancer cell lines of this study. Figure 3A summarizes the 20 over-represented REACTOME terms which are significantly associated with the upregulated genes. This analysis suggested that the upregulated DEGs were most significantly categorized into functional groups related to circadian regulation, group of pathways related to phosphorylated eIF2-alpha translation factor, FOXO-mediated transcription, metabolism of diverse lipids, events of membrane trafficking, its regulation by RAB GTPases and UPR, transport of amino acids and inorganic cations, biogenesis of mitochondria, apoptosis mediated by BH3-only proteins; strikingly, the activation of the signaling by receptor tyrosine kinases, which is the route that Sfb mainly inhibits, is among the categories that are also upregulated; this might be interpreted as a cellular response to the Sfb inhibition in order to regain cellular homeostasis. Figure 3B shows the 20 over-represented REACTOME terms which are downregulated by Sfb. These mainly include the routes of synthesis of cholesterol and steroid, DNA replication, metabolism of RNA, and cell cycle. Other processes such as translation and formation of the mitochondrial respiratory complex I were also downregulated; again, as a possible adaptation mechanism, Sfb upregulates the transcriptional activation of mitochondrial biogenesis (Figure 3A). Thus, we conclude that Sfb both induces the activation of the expression of genes belonging to pathways involved in signal transduction, that regulate cell metabolism, belong to the integrated stress response and trigger apoptosis and negatively affects the expression of genes mainly involved in cell growth, cell cycle control, and proliferation.

### 3.2. Sorafenib Inhibits Cell Growth

Several studies have clearly shown that treatment with Sfb reduced cell viability and promoted cell death in HCC cell lines (e.g., [13] and references therein). Consistently, Figure 4A shows a substantial decrease in cell proliferation upon the treatment of Sfb in our growing conditions, which is compatible with the cytotoxic activity of this drug. Interestingly, our RNA-Seq analysis found, among the different functional groups of downregulated DEGs, those related to cell growth, ribosome biogenesis, translation, co-translation protein targeting to membrane, cell cycle DNA replication and repair, cell cycle checkpoints, and sterol biosynthesis. All showed high negative values of normalized enrichment score (NES) in the GSEA analysis, thus reflecting a significant enrichment of these pathways in the list of predominantly inhibited genes upon Sfb treatment in HepG2 cells (Appendix A and Figure 3A).

Regarding the cell cycle and DNA replication categories, we found that the levels of different E2F isoforms and of the kinase CDK4 were significantly and mildly downregulated, respectively. In agreement with a reduction in the function of different E2F isoforms, the most severe one of E2F2, Sfb causes a clear delay at the S and G2 phases of the cell cycle in HepG2 cells (Figure 4B). It should be emphasized that a selective CDK4/6 inhibitor, Palbociclib, has shown promising results by impairing tumor growth in vivo and significantly increasing patients’ survival in various preclinical models of retinoblastoma-positive HCC when used alone or particularly in combination with Sfb [35].

Regarding DNA replication and repair, it is worth mentioning that downregulation of gene expression by Sfb involves many components of the minichromosome-maintenance proteins (MCM) complex, as previously reported by microarrays [10]. Expression of genes coding for proteins of the DNA damage sensor such as ATR and its stabilizer ATRIP, Claspin and RPA is also downregulated; all these proteins are involved in a checkpoint leading to the inhibition of the firing of the DNA replication origins as a consequence of DNA damage due to genotoxic stress (e.g., ionizing radiation, ultraviolet light) or a DNA replication stall accident (e.g., [36,37]). Figure 5 summarizes these findings. We also found downregulation of genes coding for proteins belonging to the base excision repair pathway, including those coding for DNA glycosylases such as UNG, SMUG1 and NEIL1, the endonuclease FEN1 and subunits of DNA polymerases delta and epsilon [38]. As previously discussed [10], the putative inhibition of DNA replication and repair pathways upon Sfb treatment would make HCC cells more sensitive to chemotherapeutic drugs inducing DNA damage (e.g., doxorubicin), irradiation or drugs suppressing DNA damage repair signaling (e.g., olaparib, THZ531), making pertinent the combination therapy as of clinical utility [39,40,41,42]. Finally, the telomere replication seems also to be negatively affected by Sfb, as shown by the fact that two functional categories related to this process are among the Sfb-downregulated pathways in our study. Telomeres have also been rationalized as useful targets in hepatocarcinogenesis [43].

All in all, the most affected REACTOME categories by Sfb are related to all steps of the ribosome biogenesis, protein synthesis, protein transport, and protein homeostasis (Figure 5). Thus, Sfb treatment produced an important reduction in the expression of practically all genes encoding cytosolic and mitochondrial ribosomal proteins, several subunits of RNA polymerases I and III and ribosome assembly factors of both 40S and 60S ribosomal subunits. Sfb also reduced the expression of genes encoding different initiation and elongation translation factors and components of the signal recognition particle (SRP)-dependent co-translational protein targeting the ER lumen, including the translocon complex. Interestingly, Sfb also visibly impacted on the proteasome-dependent protein degradation pathway, as the expression of many genes encoding proteasome regulators and components is also clearly reduced. In agreement with the role of Sfb perturbing protein homeostasis, we and others have demonstrated that Sfb at concentrations achievable in patients, induces ER stress characterized by a concomitant increase in the phosphorylation status of eIF2alpha, reduction in the phosphorylated form of mTOR, and an inhibition of protein synthesis at the initiation phase in HepG2 cells [13,44,45]. In agreement, the REACTOME categories of ATF4 activates genes and PERK regulation are among the most prominent upregulated processes upon Sfb treatment and the gene encoding the CPEB4 protein is also upregulated (Figure 3A and Appendix A). Herein, we confirm that Sfb also inhibits translation initiation in SNU423 cells, making its effect general through this process (Appendix A). This inhibition is not complete, indeed, the co-administration of eIF4E and eIF4G inhibitors with Sfb enhanced the negative effects of the latter on cell growth, viability and induced more early apoptosis in liver cancer cell lines HepG2 and Huh7 [46]. Regarding proteasome-related degradation pathways, as aforementioned, treatment of HepG2 cells with Sfb causes a reduction in the expression of different genes related to this protein complex, which could also explain the proliferation inhibition and the apoptosis induction that occur in Sfb-treated cells. Consistent with this fact, we also observed an increase in the intracellular amount of ubiquitinated proteins in HepG2 cells, which is comparable to that obtained upon a treatment with proteasome inhibitors such as ALLN or Epoxomicin (J.M.; unpublished results). Interestingly, the combination of Sfb with any of these proteasome inhibitors, Bortezomib or Carfilzomib, exhibits synergistic antitumor activity against HCC, providing a potential therapeutic strategy to improve the clinical outcome of this fatal disease [47,48,49].

### 3.3. Sorafenib Downregulates Mitochondrial Functions, Especially the Oxidative Phosphorylation Pathway

The induction of mitochondrial dysfunction and metabolic reprogramming are key events during Sfb-induced cytotoxicity in liver cancer cells (e.g., [50]). Recent data demonstrate that Sfb indeed impairs different mitochondrial functions, among them the oxidative phosphorylation pathway, in HCC cell lines [51,52,53]. All these dysfunctions are related to the impairment of complexes I, III and V in the electronic respiratory chain and mitochondrial membrane depolarization [50,52,54]. Simultaneously, Sfb stimulates glucose uptake consumption and aerobic glycolysis in glucose-grown cells and increases the amount of mitochondrial reactive oxygen species (ROS) [52]. Damaged mitochondria also burst and release cytochrome c in the cytoplasm, which eventually helps with the cell death process either by apoptosis or necrosis observed in Sfb-treated cells [55,56]. Interestingly, in our analysis we found downregulated expression for a large number of genes encoding subunits of the respiratory complex I (NADH:ubiquinone oxidoreductase), as well as for a selection of genes encoding subunits of the respiratory complexes II or IV, the pyruvate dehydrogenase or enzymes from the Krebs cycle (Figure 6A). As a possible feedback regulation, genes related to the transcriptional activation of mitochondrial metabolism are upregulated in parallel (Figure 3A). To validate these results, we determined the protein levels of different subunits of complex I of the mitochondrial respiratory chain by Western blotting. As shown in Figure 6B, a moderate but significant reduction in the levels of these proteins compared to those of the housekeeping GAPDH control was detected in extracts of HepG2 cells treated with Sfb for 12 h. Thus, Sfb impairs oxidative phosphorylation by different means, among them by downregulation of several genes encoding components of the respiratory complexes. Clearly, this downregulation of the simultaneous expression of most components of distinct respiratory complexes must be coordinated by the same regulatory factor(s) to properly control their stoichiometry. In keeping with the fact that Sfb reprograms the glucose metabolism, we also found downregulation of pyruvate dehydrogenase, an enzyme that links aerobic glycolysis to the TCA cycle and respiration (Figure 6A). In agreement with these results, it has clearly been reported that glucose withdrawal or the use of 2-deoxy-glucose, a glycolysis inhibitor, dramatically increase the Sfb toxicity of different cancer cell lines [52], the employment of Sfb in combination with a glycolysis blockade being another possibility for targeted anticancer therapy against HCC (see [57] and references therein).

### 3.4. Lipid Metabolism in Sorafenib-Treated Hepg2 Cells

It has been described that Sfb treatment disrupts lipogenesis as it reduces the expression of key lipogenic enzymes involved either in fatty acid synthesis or desaturation, including SCD1, FADS1 or FADS2 (e.g., [58]). Interestingly, selective inhibition of these enzymes has potential anticancer activity and drugs inhibiting the lipogenic process, such as Betulin or the A939572 inhibitor of SCD1, synergistically facilitate the antitumor effect of Sfb on HCC cells and/or xenograft tumors [58,59]. In our analysis, we confirmed that the Sfb treatment is associated with an upregulation of a large number of genes related to fatty acid catabolism, including different isoforms of the cytochrome P450 enzymes involved in the degradation of long-chain fatty acids [60]. Downregulation of lipogenic enzymes and fatty acid oxidation are related to AMPK activation and downregulation of the mTOR pathway [61], both mechanisms being observed in our conditions (e.g., [13]). Moreover, genes related to cholesterol biosynthesis, regulation of this pathway by SREBP and metabolism of steroids are clearly found among the genes whose expression is downregulated upon Sfb treatment (Figure 3B). These include, in HepG2 cells, genes for practically all key enzymes involved in cholesterol biosynthesis including HMGCR, MVK, PMVK, MVD, IDI1, FDPS, FDFT1, SQLE, LSS, SC5D and DHCR7 (Appendix A and Appendix A). Cholesterol synthesis has an important function supporting the growth of HCC lesions, as its upregulation is a molecular feature of aggressiveness, thus, a negative prognostic marker [62]. Testing co-administration of Sfb with cholesterol synthesis inhibitors could represent rational strategies for HCC therapy and/or even prevention.

On the other hand, our results show that Sfb upregulates the synthesis of phosphoinositol phosphates (Figure 3A), which are membrane lipids that play crucial roles in a wide range of different cellular processes, acting as second messengers in a variety of signal transduction pathways [63]. This alteration might be related to the metabolic reprogramming in lipid metabolism and carbohydrate metabolism.

### 3.5. Upregulation of the Circadian Clock in Sorafenib-Treated Hepg2 Cells

It is known that the circadian clock regulates key aspects of cell growth and survival in multiple ways, including cell cycle, senescence and metabolism (e.g., [64,65]). Liver cells have their own internal clock to temporally regulate metabolism and several genes with crucial roles in metabolic processes exhibit a circadian expression pattern [66]. Indeed, several metabolic pathways have been described as primary transcriptional targets of the hepatic clock including mitochondrial functions in respiration, protein synthesis, glucose metabolism and apoptosis, glycolysis and cholesterol metabolism [67]. In our study, the circadian clock was identified as the REACTOME category most significantly upregulated upon Sfb treatment, thus, suggesting that the misregulation found for other biological processes, especially UPR, metabolism and mitochondrial disfunctions, could be the consequence of the interference caused by Sfb to this regulatory circuit [68]. Further studies are required to challenge the circadian clock as an opportunity for HCC treatment.

### 3.6. Sorafenib at the Global Transcriptomic Level

Both HepG2 and SNU423 are epithelial liver cancer cell lines despite the well or moderate differentiation they respectively display; as such, both cell lines have high levels of E-cadherin but not increased levels of vimentin [69,70]. Recently, we have found significant differences in the antiproliferative and proapoptotic effectiveness of different drugs, including Sfb, in monolayer and spheroid cell cultures of SNU423 versus HepG2 cells [16]. To understand these differences, we compared our RNA-Seq data to identify pathways that are significantly dissimilar in HepG2 and SNU423 cell lines treated with Sfb. Thus, we selected the gene categories that expressed in the opposite way in both cell lines. Appendix A shows those biological processes that are upregulated in HepG2 while downregulated in SNU423 and vice versa. From this analysis, it can be concluded that, in agreement with what we previously deduced from Figure 1B, there are no categories showing statistical significance in the GSEA that behave in opposite directions in both cell lines. Thus, we could not find any category with a high normalized enrichment score (NES ≥ 0.9 or NES ≤ −0.9) filtering by a false discovery rate of less than 0.3. In any case, we remark the category “Rho GTPase effectors”, which is particularly upregulated in HepG2 cells while not meaningfully represented as downregulated in SNU423 cells. Rho-GTPases are a family of small signaling G proteins, whose multiple effectors are involved in a wide range of cellular processes, among them cytoskeletal dynamics for cell polarity and migration, thus, targeting Rho GTPases has become of interest to prevent cancer metastasis [71].

## 4. Conclusions

HCC is one of the deadliest human cancers. Sfb, an oral multikinase inhibitor, has been considered the standard care for patients with advanced unresectable HCC [2]. However, Sfb has only a very modest curative effect and its administration has been associated to severe adverse side effects. Moreover, a large proportion of patients develop resistance to the treatment [57]. Thus, it is challenging to develop new and effective therapies for HCC. Meanwhile, there is an obligation to discover novel ways to improve the efficacy of the Sfb treatments as a combination with other drugs that both increase its efficacy and lessen its adverse side effects.

To deepen understanding on the cellular response to Sfb, we studied its effects at a clinical reliable dosage on the global gene expression in two different model HB and HCC cell lines, HepG2 and SNU423, respectively, using RNA-Seq. This study complements those previously obtained using DNA microarray technology or available as a global GEO dataset in the literature (e.g., https://livercenter.pitt.edu/omics-data-liver-diseases; last accessed 3 February 2022). We identified DEGs in two different liver cancer cell lines. HepG2 in contrast to SNU423 cells harbord a wild-type version of p53, although this fact has not been reflected in significant differences in increasing cell proliferation or inducing apoptosis of the former versus the latter cell line [16]. Our data indicate that despite their differences in gene expression and cell dedifferentiation staging, both cell lines respond quite similarly to Sfb treatment. Thus, our analysis reveals DEGs involved in multiple biological pathways and molecular functions. Among these, we found fundamental processes such as cell cycle, DNA repair, carbohydrate and lipid metabolism, respiration and mitochondrial function, translation and the integrated stress-response, which were consistent with previous studies [10,13,54,72,73]. Some of these pathways or genes, such as the integrated stress response or autophagy, are indeed dysregulated as a way to counterbalance the cytotoxic effect of Sfb, some of which upon reactivation lead to resistances in patients treated with Sfb in monotherapy [57]. Thus, synergistic combinations of Sfb with other anticancer drugs are imperative to still use Sfb as a therapeutic strategy. Different possibilities have already been reported, unfortunately, most of them showing a very modest, if any, improved outcome [74]. Thus, drugs such as Dasatinib, Dantrolene, Trametinig, Trilostane, Copanlisib, and even Imatinib, which interfere in PI3K/AKT or ERK/MAPK have been used to enhance the action of Sfb or circumvent the Sfb-resistance (e.g., [75,76]). In turn, the combined used of Sfb with drugs such as AG-1024, Apigenin, Pristimerin or Capsaicin have been used to synergistically induce intrinsic apoptosis (e.g., [77,78]). According to our results, fundamental processes such as translation, proteasome-dependent protein degradation, cholesterol biosynthesis or RNA synthesis have promising potential to be used as targets in order to increase the efficacy of Sfb against liver cancer cells.

Few insignificant alterations in the gene expression patterns were identified as specific for HepG2 versus SNU423 cells. In our conditions, Sfb has an overall unified response, despite the fact that SNU423 are mildly more resistant to the proapoptotic and antiproliferative properties of Sfb than HepG2 cells [16].

In summary, the present study provides differential gene expression changes in an HB and an HCC cell line treated with Sfb. We not only validated previously reported findings, but also identified novel issues in terms of the basis for further research for novel therapeutic perspectives. Data presented herein will allow to enlighten previously unconnected molecular mechanisms induced by Sfb in liver cancer cells.

## Figures and Tables

**Figure 1 cancers-14-01204-f001:**
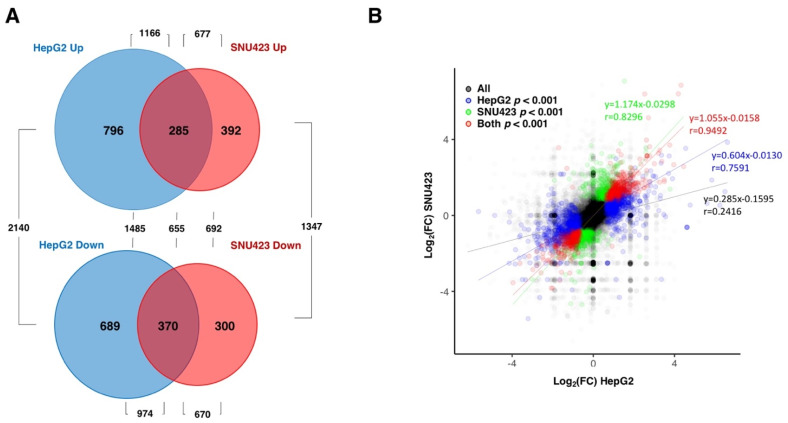
Differentially expressed genes (more than two-fold) in HepG2 and SNU423 cells upon Sfb treatment (10 µM, 12 h) detected using RNA-Seq. (**A**) Venn diagram depicting the unique and shared upregulated and downregulated genes upon treatment. The *p*-value cut-off was of 0.001. (**B**) A correlation analysis showing the relation between the RNA-Seq data of the HepG2 versus SNU423 cells upon the Sfb treatment. The regression lines for all genes, and those with *p*-values lower than 0.001 for HepG2, SNU423 or both cell lines are shown with the corresponding linear regression equations and Pearson coefficients.

**Figure 2 cancers-14-01204-f002:**
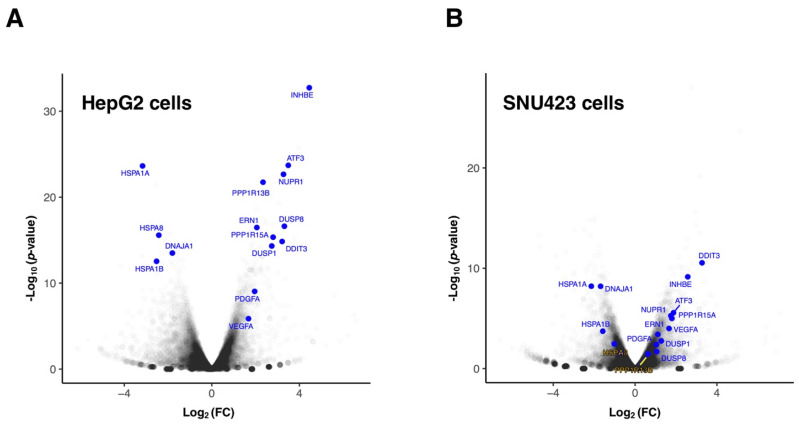
Differentially expressed genes in HepG2 (**A**) and SNU423 (**B**) cells upon Sorafenib treatment (10 µM, 12 h) detected using RNA-seq. The Volcano plot illustrates selected transcripts that show significant misregulation. The *p*-values are plotted in a −log_10_ format as y-values and fold change of the mRNAs are plotted in a log_2_ format as x-values. Representative transcripts that showed significant differential expression are indicated (blue). The rest of genes are shown in grey; density of genes is represented by a darker grey intensity.

**Figure 3 cancers-14-01204-f003:**
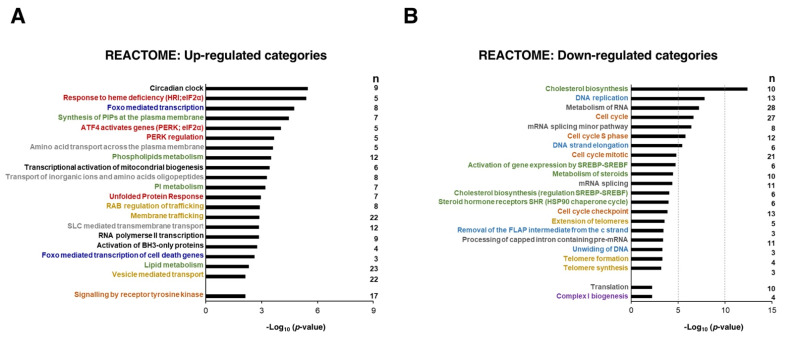
Overrepresentation analysis (ORA) of commonly upregulated (**A**) and downregulated (**B**) genes in both HepG2 and SNU423 cells treated with Sorafenib (10 µM, 12 h) using the REACTOME database. The 20 categories with better *p*-values as well as a few selected ones (e.g., signaling by receptor tyrosine kinase, translation, complex I biogenesis) are shown. The categories of related biological processes are labeled in the same color. The number of genes with *p*-value < 0.001 in the different categories (*n*) is depicted. The *p*-values were −log_10_ transformed.

**Figure 4 cancers-14-01204-f004:**
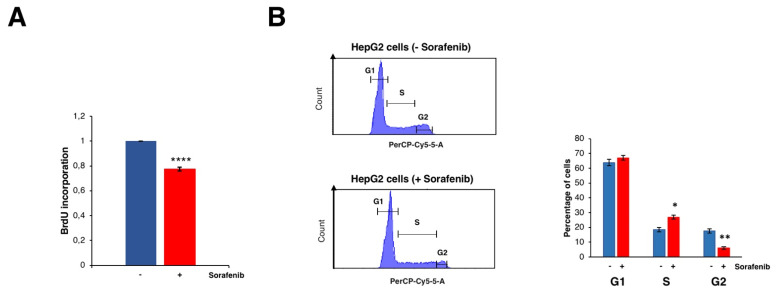
Effects of Sorafenib on cell proliferation and cell cycle. (**A**) BrdU incorporation assay on HepG2 cells treated or not with 10 µM Sorafenib. Results are expressed as the mean ± S.D. of three independent experiments. Statistical significance was analyzed using Student’s test (**** *p* < 0.0001). (**B**) Cell cycle profiles obtained by FACS of HepG2 cells treated or not with 10 µM Sorafenib. For measuring DNA content, cells were stained with propidium iodide. G1, S and G2/M phases are indicated. The percentage of cells at the different phases are shown in the histogram. Results are expressed as the mean ± S.D. values of three independent experiments. Statistical significance was analyzed using Student’s test (* *p* < 0.05; ** *p* < 0.01).

**Figure 5 cancers-14-01204-f005:**
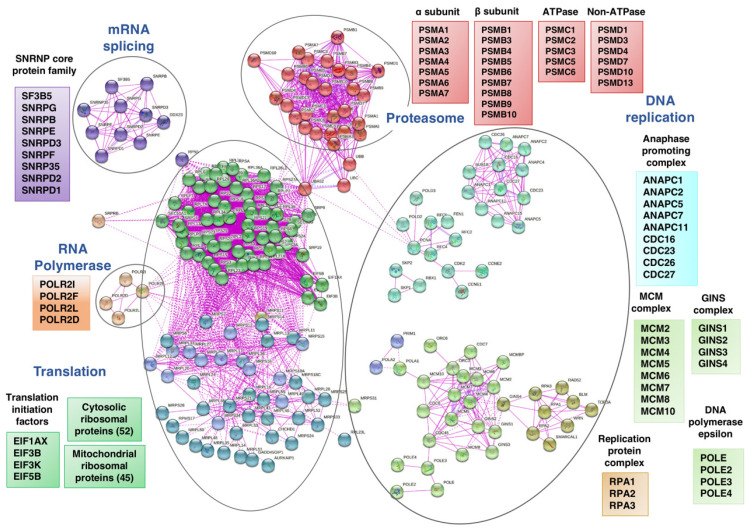
Network analysis of selected REACTOME categories of proteins corresponding to downregulated genes in HepG2 cells treated with Sorafenib (10 µM, 12 h). The analysis was undertaken using the STRING-db protein-protein interaction database (htps:/string-db.org; last accessed 3 February 2022), that relies on experimentally derived structural evidence and an interaction score ≥0.7 (highly confident entries) was used.

**Figure 6 cancers-14-01204-f006:**
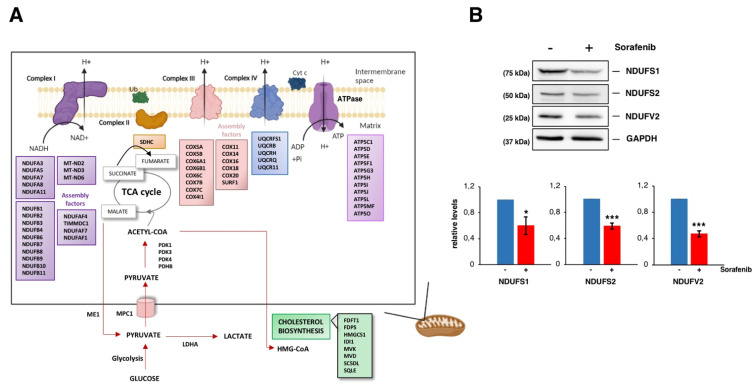
Sorafenib targets mitochondrial functions. (**A**) Cartoon showing mitochondrial-related proteins whose genes are downregulated upon Sorafenib treatment (10 µM, 12 h) in HepG2 cells. Those for genes downregulated in the cholesterol biosynthesis are also depicted. (**B**) Steady-state levels of three different core subunits of the mitochondrial membrane respiratory chain NADH dehydrogenase (Complex I) from HepG2 cells treated or not with Sorafenib (10 µM, 12 h) analyzed by Western blot with specific antibodies. The histograms show the relative expression levels of the different proteins as mean ± S.D. values of three independent experiments replicated twice. Statistical significances were analyzed by Student’s test (* *p* < 0.05; *** *p* < 0.001). The uncropped blots are shown in Appendix A.

## Data Availability

The data presented in this study are available in this article (and in the Appendix A).

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
