# Peer review of "Broad Transcriptomic Impact of Sorafenib and Its Relation to the Antitumoral Properties in Liver Cancer Cells"

_cancers, 2022, doi:10.3390/cancers14051204_

Round 1
Reviewer 1 Report
The authors made sufficient changes based on the suggestions provided, and they explained the reasons for their choices.
In the current version, the text of the manuscript is clearer.
Reviewer 2 Report
The authors have convincingly addressed the issues raised by the reviewer.
This manuscript is a resubmission of an earlier submission. The following is a list of the peer review reports and author responses from that submission.
Round 1
Reviewer 1 Report
In the present manuscript, Contreras et al. investigated the molecular mechanisms underlying the effects of Sorafenib (Sfb) on human hepatocellular carcinoma (HCC), a highly aggressive tumor of the liver. For this purpose, the authors performed a transcriptomic analysis of HepG2 and SNU423 derived cell lines following Sfb treatment at a pharmacological dose. The analysis revealed similar alterations of gene expression in the two cell lines. Specifically, genes related to membrane trafficking, stress-responsible and unfolded protein responses, circadian clock, and activation of apoptosis were predominantly upregulated, whereas the downregulation of genes involved in cell growth and cycle, DNA replication and repair, ribosome biogenesis, translation initiation, and proteostasis occurred following Sfb treatment.
The study by Contreras et al. is well-written, interesting, novel, and provides relevant insights into the mechanisms of action of Sfb at the molecular level. In addition, these data might be beneficial for novel therapeutic approaches for liver cancer treatment consisting of Sfb administration. The experiments were properly performed, the data are robust and support the conclusions drawn. Figures are designed adequately and are easy to understand. I have no major concerns about the manuscript.
Minor issue:
- The HepG2 cell line is a hepatoblastoma and not an HCC cell line. Presumably, the authors have used this cell line due to the absence of p53 wild-type human HCC cell lines and the fact that HepG2 at the molecular level is very similar to HCC cell lines. This issue should be mentioned in the text.
Author Response
Reply to the review report (Reviewer 1)
Minor issue:
The HepG2 cell line is a hepatoblastoma and not an HCC cell line. Presumably, the authors have used this cell line due to the absence of p53 wild-type human HCC cell lines and the fact that HepG2 at the molecular level is very similar to HCC cell lines. This issue should be mentioned in the text.
RESPONSE: We thank the reviewer for this observation. Although the original patent states that HepG2 derived from the liver tissue of a 15-year-old white male with a well-differentiated hepatocellular carcinoma (information incorporated into the HB-8065 ATCC reference), the Lopez-Terrada et al. Hum. Pathol. 2009, 40: 1512-1515 (reference #17 in our revised manuscript) publication nicely shows that HepG2 cell line genetically derived from hepatoblastoma. Thus, accordingly, we have modified our manuscript to indicate this (see red text in the revised manuscript).
Additionally, we have provided the publication Park et al. Int. J. Cancer 1995, 62: 276-282 (reference #18 in our revised manuscript) when the SNU423 cell line of hepatocarcinoma is firstly described in our manuscript.
Reviewer 2 Report
Major comments:
- A relevant issue concerns the exposure time of the cells to sorafenib: the cells were treated with sorafenib for a very short time (12 hours); then the mRNA was extracted. The IC50 of sorafenib in HepG2 cells has been shown to be about 10 mM after 48 hours of exposure (see for instance DOI: 10.1158/1535-7163.MCT-12-0093). Thus, it may be argued that such a short time of exposure could have not induced a stress condition capable of highlighting the differences among the two hepatocellular cancer cell lines in a compelling way, despite their established differences at molecular level (e.g. p53 status etc.). This could be the reason for the very low fold-variations observed and thus of the difficulty in clarifying findings of this transcriptomic analysis. In fact, no clear conclusion has been provided.
- A second issue concerns the selection of genes for the internal RT-PCR validation. The authors validated only 7 genes in RT-PCR since 2 genes are housekeeping (ACTB and 28S) even if it seems that ACTB has been considered one of the candidate gene. However, 7 genes were analysed in HepG2 (including ACTB) and 6 in SNU423 with only 3 genes overlapping (i.e. DUSP1, EIF4A2, TPI). The reason is not clear. In addition, such internal validation was not successful. In fact, 3 out of 7 and 4 out of 6 in HepG2 and SNU423, respectively, went in the opposite direction. Thus, please, explain the criterion used to select these genes but, more importantly, please select (possibly in a random way) 10-12 genes among those up- and down-regulated in both cell lines and perform a new validation fundamental to establish the goodness of the transcriptomic data.
3) Figure 3 is not clear, a bit chaotic: it is not clear what it includes (common altered pathways between the two cell lines? Some pathways are altered in one line, others in the other?). Please, review it to make it clearer.
Minor comments:
- Better highlight the objectives of the study, reporting them briefly at the end of the introduction
- Better clarify the meaning of the results in the conclusions.
- Carefully review the English writing through all the manuscript, especially in the conclusions.
- In the supplementary materials, in the figure of the validation, at the column of the RT-PCR, please indicate that it is a fold-variation; in addition, indicate either in the table or in the graphs the housekeeping used.
Author Response
Reply to the review report (Reviewer 2)
A relevant issue concerns the exposure time of the cells to sorafenib: the cells were treated with sorafenib for a very short time (12 hours); then the mRNA was extracted. The IC50 of sorafenib in HepG2 cells has been shown to be about 10 mM after 48 hours of exposure (see for instance DOI: 10.1158/1535-7163.MCT-12-0093). Thus, it may be argued that such a short time of exposure could have not induced a stress condition capable of highlighting the differences among the two hepatocellular cancer cell lines in a compelling way, despite their established differences at molecular level (e.g. p53 status etc.). This could be the reason for the very low fold-variations observed and thus of the difficulty in clarifying findings of this transcriptomic analysis. In fact, no clear conclusion has been provided.
RESPONSE: Thank you very much for your comments. The manuscript the reviewer indicates (Coriat et al. Mol. Cancer Ther. 2012, 11:2284-2293) is a very interesting paper showing that IC50 to Sorafenib is 15 µM (see Figure 4B, MW Sorafenib 637). We have also observed that IC50 is ranging 7,5 µM (data not shown). In fact, routinely dose used in the most part of papers ranges 5-10 µM. In addition, the selected time (12 h) is the exact point in which Sorafenib induced a shift from autophagy to apoptosis in HepG2 cell, as we have previously reported (Rodríguez-Hernández et al. J. Cell Physiol. 2018, 234:692-708).
A second issue concerns the selection of genes for the internal RT-PCR validation. The authors validated only 7 genes in RT-PCR since 2 genes are housekeeping (ACTB and 28S) even if it seems that ACTB has been considered one of the candidate gene. However, 7 genes were analysed in HepG2 (including ACTB) and 6 in SNU423 with only 3 genes overlapping (i.e. DUSP1, EIF4A2, TPI). The reason is not clear. In addition, such internal validation was not successful. In fact, 3 out of 7 and 4 out of 6 in HepG2 and SNU423, respectively, went in the opposite direction. Thus, please, explain the criterion used to select these genes but, more importantly, please select (possibly in a random way) 10-12 genes among those up- and down-regulated in both cell lines and perform a new validation fundamental to establish the goodness of the transcriptomic data.
RESPONSE: Thank you very much for your comments. We have validated more genes in a random way, exactly 15 (8 up- and 7 down-regulated) in HepG2 cells and 13 (6 up- and 7 down-regulated) in SNU423 cells and shown the data in current Figures S1 and S2. In all cases, the reference gene is 28S rDNA as indicated in the figure legend. All validations were successful and in complete agreement with our RNA-Seq data.
Figure 3 is not clear, a bit chaotic: it is not clear what it includes (common altered pathways between the two cell lines? Some pathways are altered in one line, others in the other?). Please, review it to make it clearer.
RESPONSE: We politely disagree with the reviewer. As indicated in the legend to Figure 3 and the text, the figure shows commonlyup-regulated (panel A) and down-regulated (panel B) genes in both HepG2 and SNU423 cells lines. To make the argument clearer we have slightly modified the figure legend.
Minor comments:
- Better highlight the objectives of the study, reporting them briefly at the end of the introduction.
RESPONSE: The objectives of this study have been clearly highlighted in our manuscript at the end of the introduction.
- Better clarify the meaning of the results in the conclusions.
RESPONSE: The meaning of the results and their future perspectives have been clearly highlighted in our manuscript at the conclusion section.
- Carefully review the English writing through all the manuscript, especially in the conclusions.
RESPONSE: Thanks. We have proofread the text.
- In the supplementary materials, in the figure of the validation, at the column of the RT-PCR, please indicate that it is a fold-variation; in addition, indicate either in the table or in the graphs the housekeeping used.
RESPONSE: Thanks. We have modified the figure legend to make clear that values are mRNA fold changes. Data are normalized to the values of 28S rRNA. The mRNA levels in control non-treated cells were set arbitrarily at 1.0 and the mRNA levels in treated cells with Sorafenib were relativized accordingly.